

# Technical Note: Extending the SWAT model to transport chemicals through tile and groundwater flow

Hendrik Rathjens[1], Jens Kiesel[1], Michael Winchell[1], Jeffrey Arnold[2], Robin Sur[3],

[1]Stone Environmental, 535 Stone Cutters Way, 05602 Montpelier (VT), USA
[2]USDA-ARS, Grassland Soil and Water Research Laboratory, 808 East Blackland Rd., 76502 Temple (TX), USA
[3]Bayer AG, Research & Development Crop Science, Environmental Safety Ass. & Strategy, Building 6692 2.14, 40789 Monheim, Germany

*Correspondence to*: Jens Kiesel (jkiesel@stone-env.com)

**Abstract.** The SWAT model is frequently used to simulate the transport of water-soluble chemicals in the environment such as pesticides and their metabolites originating from agricultural applications. However, the model does not simulate the transport of chemicals through subsurface tile drains and groundwater. This limitation is particularly significant in lowland regions and when simulating stable chemicals that can leach to and accumulate in groundwater. To fill this gap, the publicly

available SWAT code was modified to complement the simulation of chemicals by adding transport capabilities through tile and groundwater flow. The extended model was tested in two agricultural catchments with a typically used pesticide and one of its metabolites. Results show that the transport of the pesticide is mainly governed by surface runoff and that shallow surface tile flow contributions can be significant. Metabolite concentrations in streamflow are, however, driven by a complex spatial-temporal interplay of all surface and subsurface transport components. This highlights the advantages of applying the modified

code in catchment-scale environmental exposure studies and for developing best management practices or mitigation strategies. The new code is made available as an electronic supplement to this technical note.

## 1 Introduction

Pesticide modelling on the watershed scale has evolved to significantly support the understanding of pesticide origin and transport pathways. It also is the only approach to analyse 'what-if' scenarios for assessing anthropogenic pesticide inputs

(Arabi et al., 2008), best management practices (Zhang and Zhang, 2011), and mitigation strategies to reduce pesticide concentrations in the environment (Holvoet et al., 2007). Many models have been developed that are considered appropriate for watershed-scale simulation of pesticides (Quilbe et al., 2006), of which the Soil and Water Assessment Tool (SWAT) was selected as one of three that were most suitable. SWAT (Arnold et al., 1998), a semi-distributed model, is well known for a wide range of hydrologic and water quality applications in catchments encompassing very small to very large areas world-

wide (Gassman et al., 2007, Gassman et al., 2014). The use of the SWAT model in simulating pesticide transport at the





catchment-scale has been reported in the literature since at least 2005. An overview of peer-reviewed publications is provided in Winchell et al. (2018) who list studies conducted in North America (e.g., Vazquez-Amabile et al. 2006), Europe (e.g., Fohrer et al., 2014), and Asia (e.g., Bannwarth et al., 2014).

The standard SWAT model routes nutrient through all flow components. However, it simulates pesticide contributions to streams from surface runoff and erosion, as well as lateral subsurface flow only, and does not account for transport through subsurface tile drains and groundwater flow to streams. This simplification was made as many pesticides are mobilized by surface runoff and erosion, decayed on foliage and in soil, or moved out of the soil profile via lateral flow before they are expected to contribute to tile and groundwater flow. However, those additional transport pathways are relevant when simulating more stable, soluble, and mobile pesticides or metabolites (or other stable and mobile constituents and tracers), when working in environments with a strong interaction between surface water and groundwater, in regions with shallow groundwater tables, and in areas where tile drains exist.

The model extension presented here fills this gap by considering pesticide transport through tiles and groundwater return flow to surface waterbodies in SWAT.

## 2. Software description

### 2.1. SWAT model structure

The structure of the SWAT model allows for the prediction of flow, sediment, nutrients, and pesticide fluxes at multiple scales and locations throughout a watershed. SWAT divides the catchment into multiple subbasins. Subbasin delineation is based on the size of the catchment and the density of the stream network. Subbasins are created when two streams merge and can manually be added to represent each point where model predictions are required. A subbasin is divided into multiple hydrologic response units (HRUs), which are representative of unique combinations of land use, soils, and slope within a subbasin. Each HRU is considered an independent land unit within SWAT, and each HRU can have different parameterizations and agronomic practices, including tile drains. Within the soil layers of an HRU, fluxes are distinguished into surface runoff, lateral flow, and tile flow (if tile drains are present) that contribute to streams. Additionally, evaporation and recharge to groundwater occur. In up to two groundwater aquifers, the incoming water is partitioned into capillary rise (re-entering the soil profile from the shallow groundwater layer), storage, percolation, or outflow to streams. The second aquifer can either be unconnected (fluxes are lost from the system) or connected (fluxes are routed back to the streams). All outgoing HRU fluxes enter the streams at the upstream end of the segment and are routed to the downstream end, during which in-stream processes such as attenuation, partitioning, and degradation take place. The timing of HRU-level fluxes entering an associated stream is a function of the subbasin time of concentration and does not vary across HRUs. The process of a parent chemical forming a metabolite is not implemented in the current SWAT version. Thus, the metabolite formation requires a separate calculation and implementation in the model using 'pseudo' chemical applications. For further information on the calculation of fluxes and concentrations of constituents, the reader is referred to the SWAT theoretical documentation (Neitsch et al., 2011).





## 2.2. Description of the new subsurface transport functionality

SWAT's source code (publicly available at https://bitbucket.org/blacklandgrasslandmodels/swat_development; this study is
based on version 681) consists of around 300 individual Fortran subroutines in which the processes and the functional
modelling workflow are implemented. This modular structure allows the implementation of new functionalities through adding
new routines or the modification of single subroutines.

Figure 1 shows a schematic representation of the newly implemented SWAT pesticide routing scheme. The current version of
the SWAT model prevents soluble pesticide from flowing through tile drains or entering the groundwater. While the tile and
groundwater flow along with several loadings (e.g., nutrients) are simulated, the pesticide load is not routed. Thus, subroutines
simulating subsurface flow and transport processes were modified to enable pesticide flux together with the tile flow and
groundwater flows. The tile drain pesticide routing calculations were implemented in the pesticide leaching routine (*pestlch.f*)
and a newly introduced subroutine (*pestgw.f*) contains the algorithms to simulate pesticide transport via shallow and deep
groundwater flow.

## 75  2.3. Tile drain flow pesticide implementation

Vertical and lateral pesticide movement within the soil layer as well as percolation out of the soil layer is calculated in
subroutine *pestlch.f*. For adding the capability of routing pesticide through tile drains, corresponding equations were added to
this subroutine. First, if tile drains are implemented in the respective soil layer, tile flow is added to the water that is leaving
the layer (Eq. (1)):


$$q_{lyr} = q_{tile} + q_{prk} + q_{lat} \tag{1}$$

where $q_{lyr}$ is the pesticide transport-effective flow leaving the tile-drained soil layer (without evaporation or plant uptake)
(mm H2O), $q_{tile}$ is the tile flow leaving the soil layer (mm H2O), $q_{prk}$ is the percolation out of the layer (mm H2O), and $q_{lat}$
is the lateral flow leaving the soil layer (mm H2O). Based on $q_{lyr}$, the amount of soluble pesticide leaving the tile-drained soil
layer is calculated with Eq. (2) for each pesticide (see Neitsch et al. (2011) Chapter 4:3 and Leonard et al. (1987)):

$$pst_{rem} = sol_{pst} \cdot \left( 1 - e^{-\frac{-q_{lyr}}{w_{sat} + f_{ads} \cdot sol_{bd} \cdot sol_d}} \right) \tag{2}$$

where $pst_{rem}$ is the total amount of soluble pesticide removed from the soil layer (kg ha⁻¹), $sol_{pst}$ is the initial amount of total
pesticide in the soil layer (kg ha⁻¹), $w_{sat}$ is the amount of water in the soil layer at saturation (mm H2O), $f_{ads}$ is the soil
adsorption coefficient (mg kg⁻¹ / mg L⁻¹), $sol_{bd}$ is the soil bulk density of the soil layer (mg m⁻³), and $sol_d$ the depth of the soil





layer (mm). Pesticide concentration in tile flow is then calculated by dividing the amount of removed pesticide through the flow leaving the layer (Eq (3)):


$$pst_{conc} = \frac{pst_{rem}}{q_{tile}+q_{prk}+q_{lat}}$$ (3)

where $pst_{conc}$ is the concentration of pesticide in the water (including the tile flow) leaving the soil layer (kg ha$^{-1}$ mm$^{-1}$). Finally, Eq (4) is then:


$$pst_{tile} = pst_{conc} \cdot q_{tile}$$ (4)

where $pst_{tile}$ is the amount of pesticide leaving the soil layer through tile flow (kg ha$^{-1}$).

**2.4.    Groundwater flow pesticide implementation**

The newly introduced subroutine *pestgw.f* contains equations to calculate pesticide transport via groundwater. First, the amount of pesticide in the shallow aquifer is calculated with Eq. (5):

$$pst_{rchrg} = \left(1 - e^{-1.0/gw_{delay}}\right) \cdot pst_{sol} + e^{-1.0/gw_{delay}} \cdot pst_{rchrg-1}$$ (5)


where $pst_{rchrg}$ is the amount of pesticide entering the shallow aquifer (mg ha$^{-1}$), $gw_{delay}$ is the time required for water and its soluble loadings to reach the shallow aquifer from the bottom of the root zone (days), $pst_{sol}$ is the daily amount of pesticide leached from the soil profile (mg ha$^{-1}$), $pst_{rchrg-1}$ is the amount of pesticide entering the shallow aquifer on the previous day (mg ha$^{-1}$). The current pesticide mass is tracked with Eq (6):


$$pst_{shallst} = pst_{shallst-1} + pst_{rchrg}$$ (6)

where $pst_{shallst}$ and $pst_{shallst-1}$ is the amount of pesticide stored in the shallow aquifer on the present and previous day (kg ha$^{-1}$), respectively. Then, the pesticide groundwater contribution to streamflow is calculated with:


$$pst_{shallconc} = \frac{pst_{shallst}}{d_{shall} \cdot f_{pstgw}+q_{gwshall}+revap+q_{gwseep}}$$ (7)

$$pst_{gwshall} = pst_{shallconc} \cdot q_{gwshall}$$ (8)





where $pst_{shallconc}$ is the pesticide concentration in shallow aquifer groundwater (mg ha$^{-1}$ mm$^{-1}$), $d_{shall}$ is the depth of water

in the shallow aquifer (mm H2O), $f_{pstgw}$ is the shallow groundwater pesticide mixing factor (dimensionless), $q_{gwshall}$ is the

shallow groundwater contribution to streamflow, $revap$ is the amount of water moving from the shallow aquifer into the soil

profile or being taken up by plant roots in the shallow aquifer (mm H2O), $q_{gwseep}$ is the amount of water recharging the deep

aquifer, and $pst_{gwshall}$ is the amount of pesticide entering the channel via shallow aquifer groundwater flow (kg ha$^{-1}$). The

amount of pesticide in the shallow aquifer is then updated with:


$$pst_{gwseep} = pst_{shallconc} \cdot q_{gwseep} \tag{9}$$

$$pst_{shallst} = pst_{shallst-1} - pst_{gwshall} - pst_{gwseep} \tag{10}$$

where $pst_{gwseep}$ is the amount of pesticide recharging into the deep aquifer (kg ha$^{-1}$). The deep aquifer pesticide contribution

to streamflow is then calculated with:

$$pst_{deepst} = pst_{deepst-1} + pst_{gwseep} \tag{11}$$

$$pst_{deepconc} = \frac{pst_{deepst}}{d_{deep} \cdot f_{pstgwdeep} + q_{gwdeep}} \tag{12}$$

$$pst_{gwdeep} = pst_{deepconc} \cdot q_{gwdeep} \tag{13}$$


where $pst_{deepst}$ and $pst_{deepst-1}$ is the amount of pesticide stored in the deep aquifer on the present and previous day (kg ha$^{-1}$),

respectively, $pst_{deepconc}$ is the pesticide concentration in deep aquifer groundwater (mg ha$^{-1}$ mm$^{-1}$), $d_{deep}$ is the depth of water

in the deep aquifer (mm H2O), $f_{pstgwdeep}$ is the deep groundwater pesticide mixing factor (dimensionless), $q_{gwdeep}$ is the

deep groundwater contribution to streamflow, and $pst_{gwdeep}$ is the amount of pesticide entering the channel via deep aquifer

groundwater flow (kg ha$^{-1}$). Finally, the pesticide amount in the deep aquifer is updated:

$$pst_{deepst} = pst_{deepst-1} - pst_{gwdeep} \tag{14}$$

In addition, minor changes were made to other subroutines for technical reasons, e.g., to produce HRU level output, track

pesticide fluxes in all flow components, and write the fluxes to output files. These changes are not discussed here but are

included in the code provided in the electronic supplements.

The model will be largely compatible with the input files of the original SWAT code. The only change required to the default

SWAT input parameters is the addition of the groundwater mixing parameters in the basins.bsn input file. These two

parameters must be added to line 136 and 137 of the basins.bsn input file manually and have the default values of:





155       *1.0000   | PESTGWFACTOR: mixing factor of pesticide entering shallow gw aquifer - 0 no mixing, 1 complete and instantaneous mixing.*

          *1.0000   | PESTGW_D_FACTOR: mixing factor of pesticide entering deep aquifer - 0 no mixing, 1 complete and instantaneous mixing.*

A compiled Windows executable and the complete model code are provided as electronic supplements.

**3.      Application**

Application of the modified SWAT model was conducted in two agricultural catchments in Western Europe. The catchment characteristics are summarized in Table 1. Catchment names and location as well as detailed descriptions and names of the chemicals were anonymised for this publication. In both catchments, pesticide application data were available along with observations of streamflow, pesticide, and pesticide metabolite concentrations. All data sources overlap temporally from June

2016 to December 2019 for catchment 1 (C1) and from June 2010 to December 2013 for catchment 2 (C2). The parent pesticide is a commonly used chemical typically applied in late autumn on winter grains or in spring on corn. Based on the pesticide's half-life, it is classified as "readily degradable", its mobility is classified as "moderate", and it is considered "readily soluble" in water (FAO, 2000). In contrast, the metabolite is stable ("very slightly degradable"), "highly mobile", and "highly soluble". Model parameterization followed standard procedures considering information on climate, topography, soil, and land use

properties. Application data on respective crops were available with approximate amounts and timing for C1 and as field-specific applications for C2. A multi-metric calibration was conducted combining visual comparison and multiple performance metrics for both streamflow and concentration of the chemicals using the modified SWAT code. The entire record of observed chemical concentrations was selected as the calibration period and a second independent validation period was not selected. This is a common approach used for hydrologic and pesticide model calibration when the observed data period is relatively

short (Daggupati et al., 2015).

Figure 2 and Figure 3 show discharge and pesticide and metabolite concentrations (columns) for the different flow components (rows) for C1 and C2, respectively. The hydrologic calibration led to a good visual agreement between observed and simulated discharge and good to very good performance statistics with daily NSE values of 0.76, 0.63 and PBIAS of 6.6%, 1.8% in C1 and C2, respectively. The pesticide and metabolite concentrations in streamflow simulated with the original SWAT code (grey lines)

lines) and the modified code (red lines) are shown in Figure 2 and Figure 3. The original and modified SWAT pesticide simulations returned similar results and a good fit between modelled and observed concentrations was achieved. However, C1 was characterized by very few detections above the level of quantification and the highest observed pesticide peak could not be reproduced by both models. In C2, reported point source inputs of the pesticide occurred (likely due to mistreatment of the product) which are not included in the model, which lead to discrepancies between simulated and observed concentrations.

The metabolite dynamics and magnitudes cannot be reproduced by the original SWAT code in both catchments, but are very well represented by the modified code, emphasizing the importance of the subsurface transport processes for the metabolite.



The contribution of the different flow components to discharge is similar in both catchments, where surface runoff has the highest impact on peak flows, tile drains are relevant mostly in the wetter months and shallow groundwater shows a clear seasonal dynamic with lowest flow values in summer. Lateral flow and deep groundwater flow have low contributions. The
deep groundwater aquifer, however, sustains the flows in the summer periods. The modified SWAT code allows the output of concentrations in all flow components. They show a similar pattern in both catchments where surface runoff and lateral flow are the only flow components with significant concentrations. All simulated pesticide peaks can be attributed to surface runoff events as comparably high pesticide concentrations in streamflow coincide with a significant runoff event. Pesticide concentrations in lateral flow are also significant, but the contribution of lateral flow to total streamflow is low, and loadings
in lateral flow are therefore significantly diluted when entering the stream. The concentrations of the metabolite in streamflow have a comparable magnitude in C1 and C2 (maximum between 10 to 15 µg/L) and the dynamics of the concentrations in the transport components have a similar pattern in both catchments. Concentrations in lateral flow fluctuate, but are always greater than zero, indicating a constant presence of the metabolite in the soil. This can also be seen in the high tile drain concentrations that lead to substantial metabolite contributions when tile flow occurs. The metabolite is also permanently present in the
groundwater, which is a significant transport pathway.

These results show that the fast flow components are responsible for the pesticide concentrations in streamflow. For the metabolite, a single most important transport process cannot be identified and a complex interplay between multiple transport pathways is responsible for the concentration dynamics and magnitude: Lateral flow and shallow groundwater flow are the most import input pathways in summer during low flow conditions and tile drain flow during autumn and winter. Surface
runoff and deep groundwater flow have negligible contributions.

## 4.    Summary and conclusion

The SWAT model code was extended to simulate pesticide transport through tile drains and two groundwater layers. All subroutines and a compiled executable are provided in the electronic supplements to this technical note. For applying the updated model, minor changes must be made to the standard SWAT input files with two additional parameters in one input
file.

The application of the implemented code in the two case study catchments demonstrates the advantages to simulating pesticide transport through tile drains and groundwater flow. The actual concentrations in the respective transport pathways and water balance components are available over time, which are important information to assess environmental fate and transport processes. It is also apparent that the complex temporal interplay between all flow components, including tile and groundwater
flow, is needed to sufficiently simulate concentrations of metabolites and other chemicals with similar properties in streamflow. Visualizing the pesticide and metabolite concentration in all flow components improves the understanding of the origin of the chemicals. This supports a more targeted calibration of the models and provides important information to develop best management practices to mitigate potential contamination of surface and groundwater.

The developed software fills a gap in watershed scale pesticide modelling. The code refinements were made available to the SWAT development team and will potentially be included in a future official revision of the SWAT model. The next version of the SWAT model, called SWAT+, (Bieger et al., 2017), will include the simulation of pesticides in all hydrological flow pathways and a direct simulation of metabolite formation using a first order decay function. However, while scientists and watershed managers are in the process of transitioning to SWAT+ and its supporting interfaces become available, the extended SWAT version is a valuable tool for risk managers and exposure modelers.

**Code availability**


The source code and compiled Windows executables are available from Stone Environmental's GitHub repository (https://github.com/StoneEnv/SwatPestTileGw) under the GNU General Public License v3.

**Author contribution**

**Conceptualization**: Hendrik Rathjens, Michael Winchell and Robin Sur; Data curation: Jens Kiesel, Hendrik Rathjens,
Michael Winchell and Robin Sur; **Methodology**: Hendrik Rathjens, Jens Kiesel and Michael Winchell; **Software**: Jeffrey Arnold, Hendrik Rathjens, Michael Winchell; **Supervision**: Robin Sur; **Validation**: Jens Kiesel, Hendrik Rathjens, Michael Winchell; **Visualization**: Jens Kiesel; **Writing – original draft**: Jens Kiesel; **Writing – review & editing**: Jens Kiesel, Hendrik Rathjens, Michael Winchell, Jeffrey Arnold and Robin Sur.

**Competing interests**

The authors declare no competing interests.

**Disclaimer**

The authors accept no responsibility for any liability arising from the use of this manuscript, the provided source code and model.

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





**Table 1: Catchment characteristics of the two anonymized catchments in Western Europe**

| Catchment Characteristics | Unit | Catchment 1 | Catchment 2 |
|---|---|---|---|
| Catchment area at gauge | km² | 38.0 | 9.9 |
| Elevation gradient | mASL | 45-110 | 24-159 |
| Landuse distribution | - | Agriculture (73%) Forest (17%) Urban (10%) Pasture (2%) | Agriculture (80%) Pasture (13%) Forest (6%) |
| Tile drained | % | 52 | 48 |
| Average annual precipitation (min-max) * | mm | 641-809 | 631-945 |
| Average annual maximum temperature (min-max) * | °C | 13.1-15.6 | 13.3-15.4 |
| Average annual minimum temperature (min-max) * | °C | 4.3-6.1 | 5.6-7.1 |
| Mean runoff rate as percent of precipitation ** | % | 28-36 | 38-48 |
| Number of subbasins | - | 39 | 17 |
| Number of HRUs | - | 5163 | 922 |

* time period Jan-2008 to Dec-2013

** time period Jun-2010 to Dec-2013






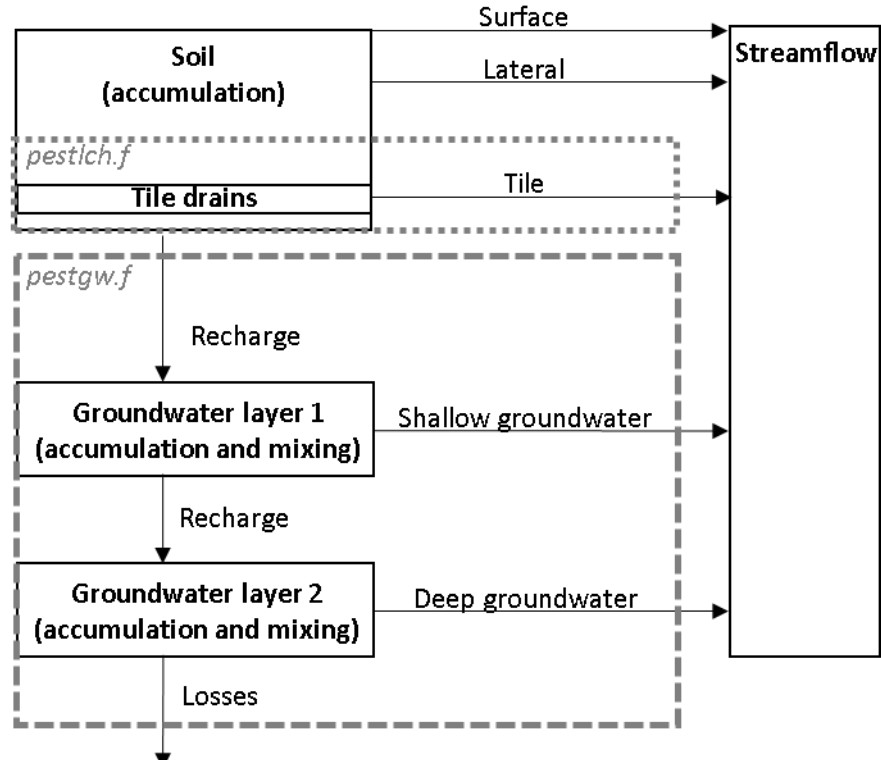

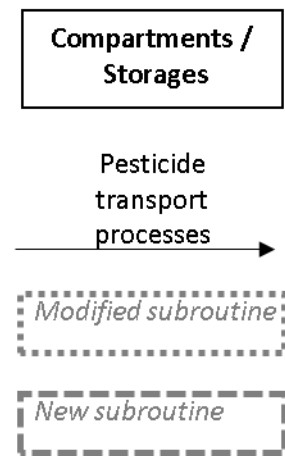

**Figure 1: Flow chart of the newly implemented pesticide routing functionality**





**Figure 2: Catchment 1 (C1) time series (Jun 2016 – Dec 2019) for observed and simulated discharge, parent pesticide, metabolite in streamflow and simulated time series for all flow components**



**Figure 3: Catchment 2 (C2) time series (Jun 2010 – Dec 2013) for observed and simulated discharge, parent pesticide, metabolite in streamflow and simulated time series for all flow components**