# Peer review of "Technical Note: Extending the SWAT model to transport chemicals through tile and groundwater flow"

_Hydrology and Earth System Sciences, 2022_

## Author Comment (AC1)

**General Comments**

In their manuscript "Extending the SWAT model to transport chemicals through tile and groundwater flow" Rathjens et al. et al. describe the implementation of chemical transport for groundwater and tile drainage in the Soil and Water Assessment Tool (SWAT). This functionality can be of interest for researchers involved in transport modelling on landscape-level, for example, for academia, authorities, environmental agencies, water suppliers and other stakeholders. Therefore, the topic is appropriate for Hydrology and Earth System Science since this kind of research contributes to the understanding of hydrology and transport in the environment.

However, some points have to be considered before the manuscript can be published. Specifically, the calibration for the exemplary studies sites and how parameterization interacts with the modifications made have to be explained in more detail. This would add value to the justification of the modifications.

Therefore, minor revisions are suggested. Please find more detailed comments in the following paragraph.

Response: Thank you for taking the time to review our manuscript and for the helpful suggestions.

**Detailed Comments**

1. Application (chapter 3; lines 171 ff): It would be desirable to get more detailed information on the calibration process and the selected parameters: Are both model parameterizations (with and without modification) similar for each test site? Please clarify.

   Response: The parameterizations for the runs with and without modifications are the same (apart from the newly introduced parameter PEST_GW_D, which is not used in the original code) but they differ for each test site (see response to next comment).
   We first calibrated the model using the modified SWAT version and then applied the original SWAT model on the same input files. With regard to the focus of the manuscript two points should be noted. (1) The shown comparison "modified" vs "original code" are meant to demonstrate that the modified routines realistically represent hydrologic transport processes. It was not our intention to provide a completely "equitable" model comparison between performance of the two model versions. (2) The newly implemented processes enable the modeler to evaluate concentrations and dynamics in each flow component, which supports process understanding and improves

process representation in the model. This is especially true for simulating transport of the metabolite, which requires a correct representation of the individual flow components. For the parent compound it might be possible to achieve acceptable fits for both model versions if calibrated individually at the gauge. However, this could lead to wrong conclusions when interpreting the model results obtained from the original version as important processes (transport through tile drain and ground water) are not considered. In addition, achieving a similar performance with the original version is not possible for the metabolite because in both watersheds the metabolite is mainly driven by ground water transport, whose dynamics differ significantly from surface and lateral flow.

We will add the following explanation to the manuscript, Section 3, l.175:

The models for the two watersheds were first calibrated using the modified code and then both model versions (original and modified) were run using the same parameters. This is not meant to be a completely 'equitable' model performance comparison, but to show the differences between the two versions.

2. What are the parameters involved in the calibration? What is their impact on groundwater and drainage contribution to streamflow? Please indicate in more detail with specific focus on code updates made.

   Response: We agree that providing a list of parameters, including the calibrated parameter values would be useful. We compiled a table (see below) of the parameters involved in the calibration. As shown therein, the calibration targeted the main hydrological processes and therefore also impacted groundwater and tile drainage contribution. The hydrologic processes regarding groundwater and tile drainage are the same in both model versions and our modifications only impacted the chemical transport processes. We are, however, unable to assess the impact of the code updates regarding pesticide groundwater and drainage contribution, as the original code does not simulate pesticide transport in those flow components. It should also be noted that this is a short communication and that a full discussion on the impact of individual parameters on flow components goes beyond the scope of the paper and article format. To address the comment, we will include Table 3 (see below) and add an extended description of the pre-calibration parametrization and the calibration process to the manuscript (Section 3):

SWAT offers a range of algorithms representing hydrological processes. Based on experience and understanding of the catchment characteristics, the Hargreaves potential evaporation method and the evaporation-based daily curve number

adjustment method were chosen. Pre-calibration-settings of hydrologic parameters included the adjustment of heat units to ensure crops develop completely and the adjustment of channel roughness to account for vegetated, small channels. Pesticide-related algorithms and parameters updated prior to calibration included pesticide in-stream processes such as burial and volatilization, which were turned off due to the low Koc, Henry Low Constant, and vapor pressure of both chemicals and the short travel time in the two catchments. In addition, metabolite release in the soil was parametrized to account for metabolite formation in the soil profile using 'pseudo' chemical applications for both model versions.

A list and description of the calibration parameters and the processes they are associated with is provided in Table 2. A parameter is included in the table if it was changed in at least one of the catchments.

Table 2: Calibration parameters with initial value and calibrated end value (changed values in bold)

| | SWAT Parameter | Parameter Description | Initial Value | Calibrated end value | |
| --- | --- | --- | --- | --- | --- |
| | | | | C1 | C2 |
| Surface runoff | CNCOEF | Plant ET curve number coefficient. | 1 | **1.1** | 1 |
| | SURLAG | Surface runoff lag coefficient. | 1 | 1 | **0.5** |
| Tile drains | DEPIMP | Depth to restrictive layer (mm) | N/A | **2010** | **2250** |
| | GDRAIN | Drain tile lag time (hr) | 0 | **2** | **12** |
| | TDRAIN | Time for tiles to drain soil to field capacity (hours). | 48 | 48 | **24** |
| | DDRAIN | depth to subsurface tile drain (mm) | 1000 | **990** | 1000 |
| Groundwater | ALPHA_BF | Baseflow alpha factor | 0.048 | **0.77** | **0.01** |
| | GWDELAY | Groundwater delay (d) | 31 | **47.4** | **1** |
| | ALPHA_BF_D | Baseflow alpha factor for deep aquifer | 0.01 | 0.01 | **0.0001** |
| | GWQMIN | Threshold depth of water in the shallow aquifer required for return flow (mm) | 1000 | 1000 | **500** |
| | RCHRG_DP | Deep aquifer percolation fraction | 0.05 | 0.05 | **0.15** |
| Soil | AWC | Available water capacity | varies by soil | **1.1*default** | **1.33*default** |
| | ESCO | Soil evaporation compensation factor | 0.95 | 0.95 | **1** |
| Pesticide and Metabolite | PERCOP | Pesticide percolation coefficient | 0.5 | 0.5 | **0.6** |
| | HLIFE_S (Pesticide) | Soil Half-Life (d) | N/A | **14** | **35.7** |
| | PESTGWFACTOR | mixing ratio of pesticide entering shallow gw aquifer (-) | 1 | 1 | **0.02** |
| | PEST_GW_D | mixing ratio of pesticide entering deep gw aquifer (-) | 1 | **0.02** | 1 |

3.  A "multi-metric" calibration is mentioned. Besides visual inspection, was there any weighing used between streamflow and concentration metrics for calibration or, resp., a multi-objective calibration?

Response: The multi-metric calibration used different metrics that focus on high flows, low flows, and different parts of the flow duration curve and the pesticide exceedance probability curve. We did not create a single objective function where streamflow and concentration metrics have different weights. Instead, we used our own judgement to balance performance on water balance, streamflow, and concentrations. We will add the following explanation to Section 3:

The calibration was conducted separately and iteratively for streamflow and pesticide concentrations (i.e., no multi-objective function combining streamflow and pesticide metrics was used).

---

## Author Comment (AC2)

This manuscript is well written and suitable for publication in HESS. Tile drains and potential groundwater flow to streams are two important pathways in many agricultural watersheds. Adding these processes to the well-established SWAT model will undoubtedly expand the model utility for watershed managers, researchers, and ag-engineers and practitioners in the farming community to develop and assess best management practices and stewardship programs that support sustainable and more environmentally friendly agriculture. Two specific comments are below.

Response: Thank you for the positive feedback, for taking the time to review our manuscript, and for the helpful suggestions.

1. Section2.4 Lines 110-115. Was the general term "soil profile" defined as the same as "root zone" or the layer between the bottom of the root zone and shallow ground water? A clear description of the "soil profile" is needed to understand Eq 5.

   Response: The soil profile, as defined in SWAT, contains multiple soil layers in which the 'root zone' is defined through an additional parameter (the rooting depth). The rooting depth can be set to a depth that is shallower or the same as the soil depth. We will add this information to the manuscript to Section 2.2:

The current version of the SWAT model does not track soluble pesticide after leaching out of the soil profile (which includes the root zone below the maximum soil depth). Thus, chemicals are prevented to flow through tile drains or enter the groundwater.

2. Section 3 Lines 165-170. The author(s) should provide the specific values of the pesticide use rate and basic environmental fate parameters such as soil half life and Koc for both parent and metabolite. It is disingenuous by only providing qualitative descriptions of "readily degradable" or "moderate", etc., unless the model can take such qualitative inputs for a simulation.

   Response: We agree that these parameters are useful to understand the model simulations. We will add the following information to Section 3:

Based on the pesticide's soil half-life (between approximately 6 and 40 days, depending on soil type; Bayer Crop Science, 2018), it is classified as "readily degradable", its mobility is classified as "moderate", and it is considered "readily soluble" in water (Koc of ~250 mL/g). In contrast, the metabolite is stable, "highly mobile" (Koc of 0 mL/g), and "highly soluble" (FAO, 2000).

The pesticide's average application rates are 221g/ha in C1 and 462g/ha in C2.

3. Section 3 Line 171. Did the "multi-metric calibration" include pesticide use and fate parameters of the parent and metabolite? if yes, what are the final calibrated values?

   Response: The calibration did not include pesticide use. Soil half-life for the pesticide was modified within the reported range that varies by soil type and is reported for the two catchments in Section 3, Table 2 (see also revisions made based on the comments of Reviewer 1):

A list and description of the calibration parameters and the processes they are associated with is provided in Table 2. A parameter is included in the table if it was changed in at least one of the catchments.

Table 2: Calibration parameters with initial value and calibrated end value (changed values in bold)

| | SWAT Parameter | Parameter Description | Initial Value | Calibrated end value | |
|---|---|---|---|---|---|
| | | | | C1 | C2 |
| Surface runoff | CNCOEF | Plant ET curve number coefficient. | 1 | **1.1** | 1 |
| | SURLAG | Surface runoff lag coefficient. | 1 | 1 | **0.5** |
| Tile drains | DEPIMP | Depth to restrictive layer (mm) | N/A | **2010** | **2250** |
| | GDRAIN | Drain tile lag time (hr) | 0 | **2** | **12** |
| | TDRAIN | Time for tiles to drain soil to field capacity (hours). | 48 | 48 | **24** |
| | DDRAIN | depth to subsurface tile drain (mm) | 1000 | **990** | 1000 |
| Groundwater | ALPHA_BF | Baseflow alpha factor | 0.048 | **0.77** | **0.01** |
| | GWDELAY | Groundwater delay (d) | 31 | **47.4** | **1** |
| | ALPHA_BF_D | Baseflow alpha factor for deep aquifer | 0.01 | 0.01 | **0.0001** |
| | GWQMIN | Threshold depth of water in the shallow aquifer required for return flow (mm) | 1000 | 1000 | **500** |
| | RCHRG_DP | Deep aquifer percolation fraction | 0.05 | 0.05 | **0.15** |
| Soil | AWC | Available water capacity | varies by soil | **1.1*default** | **1.33*default** |
| | ESCO | Soil evaporation compensation factor | 0.95 | 0.95 | **1** |
| Pesticide and Metabolite | PERCOP | Pesticide percolation coefficient | 0.5 | 0.5 | **0.6** |
| | HLIFE_S (Pesticide) | Soil Half-Life (d) | N/A | **14** | **35.7** |
| | PESTGWFACTOR | mixing ratio of pesticide entering shallow gw aquifer (-) | 1 | 1 | **0.02** |
| | PEST_GW_D | mixing ratio of pesticide entering deep gw aquifer (-) | 1 | **0.02** | 1 |

---

## Author Response (AR1)

Dear Editor,

dear Dr. Micha Werner,

point-by-point responses to the reviewer comments are documented in the two revision files that we uploaded ('Reply on RC1'

5 and 'Reply on RC2'). We have included all suggested changes from these two files into the manuscript. In summary, we made the following changes:

| Line | Reviewer | Content |
|---|---|---|
| 69-70 | R2 | from leaching out of the soil profile (which includes the root zone below the maximum soil depth). Chemicals are prevented to flow |
| 67-69 | R2 | Based on the pesticide's soil half-life (approximately 6 to 40 days, depending on soil type; Bayer Crop Science, 2018), it is classified as "readily degradable", its mobility is classified as "moderate", and it is considered "readily soluble" in water (Koc of ~250mL/g). |
| 171-182 | R1 and R2 | Model parameterization followed standard procedures considering information on climate, topography, soil, land use properties, agricultural management practices (including pesticide applications), and tile drain locations. SWAT offers a range of algorithms representing hydrological processes. Based on experience and understanding of the catchment characteristics, the Hargreaves potential evaporation method and the evaporation-based daily curve number adjustment method were chosen. Pre-calibration-settings of hydrologic parameters included the adjustment of heat units to ensure crops develop completely and the adjustment of channel roughness to account for vegetated, small channels. Application data on respective crops were available with approximate amounts and timing for C1 and as field-specific applications for C2. The pesticide's average application rates are 221g/ha in C1 and 462g/ha in C2. Metabolite release in the soil was parametrized to account for metabolite formation in the soil profile using 'pseudo' chemical applications for both model versions. Pesticide-related algorithms and parameters updated prior to calibration included pesticide in-stream processes such as burial and volatilization, which were turned off due to the low Koc, Henry Law Constant, and vapor pressure of both chemicals and the short travel time in the two catchments. |
| 184-186 | R2 | The calibration was conducted separately and iteratively for streamflow and pesticide concentrations (i.e., no multi-objective function combining streamflow and pesticide metrics was used). |
| 188-192 | R1 | The models for the two watersheds were first calibrated using the modified code and then both model versions (original and modified) were run using the same parameters. This is not meant to be a completely 'equitable' model performance comparison, but to show the differences between the two |

| | | versions. A list and description of the calibration parameters and the processes they are associated with is provided in Table 2. A parameter is included in the table if it was changed in at least one of the catchments. |
|---|---|---|
| 261 | R2 | Bayer Crop Science: Bayer Crop Science Internal Report 1, 144, 2018. |
| 296 | R1 and R2 | Table 2: Calibration parameters with initial value and calibrated end value (changed values in bold) (Table not shown here) |

10  Please excuse that the marked-up version of our Manuscript is not in "Track-Change" Mode. Instead, the new text is highlighted in blue. We hope this is ok. Let us know if not.

Thank you again for handling our manuscript!

15  With best regards

Jens